# Effect of the COVID-19 Pandemic on Surgical Outcomes for Rhegmatogenous Retinal Detachments

**DOI:** 10.3390/jcm12041522

**Published:** 2023-02-15

**Authors:** Masaharu Mizuno, Kosuke Nakajima, Aya Takahashi, Tomoka Ishida, Kazunari Hirota, Takashi Koto, Akito Hirakata, Makoto Inoue

**Affiliations:** Kyorin Eye Center, Kyorin University School of Medicine, 6-20-2 Shinkawa, Mitaka, Tokyo 186-8611, Japan

**Keywords:** COVID-19 pandemic, rhegmatogenous retinal detachment, proliferative vitreoretinopathy, scleral buckling surgery, vitrectomy

## Abstract

We reviewed the medical records of 438 eyes in 431 patients who had undergone surgeries for rhegmatogenous retinal detachments (RRD) or proliferative vitreoretinopathy (PVR ≥ Grade C) to determine whether the COVID-19 pandemic had affected outcomes. The patients were divided into 203 eyes in Group A that had undergone surgery from April to September 2020, during the pandemic, and 235 eyes in Group B that had undergone surgery from April to September 2019, before the pandemic. The pre- and postoperative visual acuity, macular detachment, type of retinal breaks, size of the RRD, and surgical outcomes were compared. The number of eyes in Group A was fewer by 14%. The incidence of men (*p* = 0.005) and PVR (*p* = 0.004) was significantly higher in Group A. Additionally, the patients in Group A were significantly younger than in Group B (*p* = 0.04). The differences in the preoperative and final visual acuity, incidence of macular detachment, posterior vitreous detachment, types of retinal breaks, and size of the RRD between the two groups were not significant. The initial reattachment rate was significantly lower at 92.6% in Group A than 98.3% in Group B (*p* = 0.004). The COVID-19 pandemic affected the surgical outcomes for RRD with higher incidences of men and PVR, younger aged patients and lower initial reattachment rates even though the final surgical outcomes were comparable.

## 1. Introduction

A rhegmatogenous retinal detachment (RRD) is a major retinal disorder with a prevalence of approximately 1/10,000 individuals [1]. If untreated, RRD can progress to an irreversible reduction in vision [2]. The two most common surgical treatments for RRDs are pars plana vitrectomy (PPV) and scleral buckling (SB) surgery. PPV has become the first choice in treating RRDs, and its use is expanding due to the advances in high-speed cutters and minimally invasive procedures resulting from advances in small-incision vitrectomy. In addition, the development of wide-angle viewing systems has contributed to the increase in primary PPV surgery for RRDs [3,4]. Nevertheless, there are a number of types of RRDs that require SB, such as RRD cases associated with oral dialysis or an atrophic hole without a posterior vitreous detachment in young patients, and RRDs complicated with atopic dermatitis [5].

Proliferative vitreoretinopathy (PVR) is a major complication of an RRD in which proliferative contractile tissues form in the vitreous and result in a tractional retinal detachment. PVR with a larger size RRD, and retinal tears in the inferior quadrants have been reported to be risk factors for the initial failure of PPV for RRDs [6]. An earlier diagnosis and treatment are necessary to prevent the progression of RRDs to PVR and total RRD [7].

The COVID-19 pandemic in 2020 had a significant impact on not only social life but also on healthcare systems [8,9]. Although Japan did not implement a lockdown, it did declare a state of emergency on three occasions that resulted in a significant change in the behavior of individuals and doctors [10]. Thus, the procedures and protocols of medical institutions were changed during the pandemic. Whether these changes affected surgical outcomes has not been determined.

Thus, the purpose of this study was to determine whether the changes instituted because of COVID-19 affected the outcomes of surgeries performed on eyes with RRD. To accomplish this, we reviewed the medical records of patients who had undergone pars plana vitrectomy or scleral buckling surgery to treat RRDs before and during the COVID-19 pandemic at our hospital.

## 2. Materials and Methods

This study was conducted in accordance with the Declaration of Helsinki, and a signed informed consent was obtained from all patients. The procedures used were approved by the Ethics Committee of Kyorin University School of Medicine (approval number #1102).

### 2.1. Subjects

The patients studied were diagnosed with RRD or PVR (grade ≥ C), had undergone their initial surgery at our hospital, and were followed for at least 6 months. The patients were divided into two groups: Group A included patients examined between April to September 2020 which was when the first state of emergency declaration was issued by the Tokyo COVID-19 pandemic group, and Group B included patients who were examined from April to September 2019, before the pandemic period. There were 200 patients in Group A and 231 patients in Group B.

### 2.2. Surgical Procedures

Vitreoretinal surgery was performed by 9 vitreoretinal surgeons at the Kyorin Hospital. The pre- and postoperative findings and the surgical procedures were obtained from medical records. The preoperative factors examined were the age, sex, lens status, best-corrected visual acuity (BCVA), an incidence of high myopia with a refractive error (spherical equivalent) of >−6.0 diopters (D) or an axial length >26.5 mm, the presence of a posterior vitreous detachment, macular detachment, undetected causative retinal breaks, the number of retinal breaks, the locations of the causative retinal breaks (upper or lower quadrants), the size of the RRD (1–4 quadrants), and presence of PVR. The background factors included prior blunt trauma, retinopathy of prematurity, association with familial exudative vitreoretinopathy, and atopic dermatitis. The surgical procedures included pars plana vitrectomy (PPV), scleral buckling (SB) surgery, or a combination of PPV and SB. The postoperative factors included the final rate of retinal reattachment, BCVA at 6 months after the last surgery, and clinical characteristics of the cases that required reoperation.

The final retinal reattachment rate was determined at 6 months after the last surgery, including the removal of silicone oil (SO) tamponade. If the SO was not removed, the eye was assumed to be still detached. We excluded cases of RD with a macular hole, RRD associated with a perforating ocular trauma, and reoperated cases after an initial surgery that was performed at another hospital. The presence of a posterior vitreous detachment was determined by the presence of glial ring floaters or intraoperative findings of the residual posterior vitreous cortex made visible by triamcinolone crystals.

Cataract surgery was performed together with PPV as needed, using the Constellation^®^ Vision System (Alcon laboratories, Fort Worth, TX, USA) for vitreous surgery and the Resight^®^ wide-angle fundus viewing system (Carl Zeiss Meditec, Dublin, CA, USA) for intraoperative observation. The PPV was performed with a 25-gauge (G) or 27G system, and tamponade was performed with air, sulfur hexafluoride (SF6), propane octafluoride (C3F8) gas, or SO as needed. The SB procedure was performed under view by indirect binocular ophthalmoscopy with cryotherapy and local buckling or encircling buckling. Subretinal fluid drainage and air or gas injection were performed as needed. Scleral buckles (#287 silicone tire, #240 silicone band, #506 silicone sponge, #511 silicone sponge, MIRA, Inc., Uxbridge, MA, USA, or LABTICIAN, Oakville, Canada) were used.

### 2.3. Statistical Analyses

The data are expressed as the means ± standard deviations. The decimal visual acuity was converted to the logarithm of the minimum angle of resolution (logMAR) visual acuity for statistical examinations. For the statistical analysis, Fisher’s exact probability tests, Wilcoxon signed rank sum tests, Mann–Whitney U tests, and logistic regression model were used with the R software (ver. R-4.2.2., The R Foundation for Statistical Computing c/o Institute for Statistics and Mathematics, Vienna, Austria). A *p* of <0.05 was considered significant.

## 3. Results

A total of 438 eyes in 431 patients, consisting of 282 men and 149 women, were studied. The mean age of the patients was 53.3 ± 14.1 years, and all had undergone surgery for an RRD. Group A had 203 eyes from 200 patients, consisting of 145 men and 55 women. Their mean age was 52.5 ± 14.0 years. Group B had 235 eyes from 231 patients consisting of 137 men and 94 women. Their mean age was 54.0 ± 14.1 years.

PPV was selected as the initial surgery for 351 eyes (80%), SB for 71 eyes (16%), and PPV + SB for 16 eyes (4%). Cataract surgery was performed on 130 eyes in Group A and 164 eyes in Group B, and PPV with 25G instruments was performed on 347 eyes and on 20 eyes with 27G instruments.

### 3.1. Preoperative Factors

Comparisons of the two groups for each of the preoperative factors showed that there was no significant difference in the preoperative BCVA (*p* = 0.65), the presence of a posterior vitreous detachment (*p* = 1), macular detachment (*p* = 1), undetected causative retinal breaks (*p* = 0.09), the number of retinal breaks (*p* = 0.54), the location of retinal breaks (*p* = 0.22), and size of the RRD (*p* = 0.54). In 18 eyes (12 eyes in Group A, and 6 eyes in Group B) in which the causative retinal breaks were not detected preoperatively, the causative retinal breaks could be determined intraoperatively. The age in Group A was significantly younger than in Group B (*p* = 0.04). The number of men was significantly greater in Group A than in Group B (*p* = 0.005). PVR was detected in 17 eyes (8%) in Group A, which was significantly greater than the 5 (2%) eyes in Group B (*p* = 0.004, Table 1).

### 3.2. Background Factors

The differences in the percentages of pseudophakia, blunt trauma, high myopia, familial exudative vitreoretinopathy, atopic dermatitis, and retinopathy of prematurity between the two groups were not significant (Table 2). The incidence of patients aged more than 55 years was significantly lower in Group A than in Group B (*p* = 0.002).

### 3.3. Technique and Surgical Outcomes

PPV was performed on 156 eyes (77%) in Group A and 195 eyes (83%) in Group B, and SB was performed on 39 eyes (19%) in Group A and 32 eyes (14%) in Group B, and the combination of both procedures (PPV + SB) was performed on 8 eyes (4%, 3%) in Groups A and B. The differences in the number of SB and PPV + SB surgeries in the two groups were not significant, and the number of PPV was lower by 20% in Group A during the pandemic. The use of a gas tamponade (SF6, *p* = 0.48, C3F8, *p* = 1, air, *p* = 0.12) or SO (*p* = 1) tamponade did not differ significantly between the two groups.

The overall initial reattachment rate for the 438 eyes was 95.7%, and it was 92.6% in Group A and 98.3% in Group B. The rate was significantly lower in Group A than in Group B (*p* = 0.004). The final reattachment rate for all eyes was 99.3%, and it was 98.5% in Group A and 100% in Group B. SO tamponade was used in four eyes in Group A and four eyes in Group B. SO was removed in five eyes within 6 months and the retina remained attached. The three eyes in Group A that were not reattached were cases in which the SO was not removed at 6 months after the last surgery even though the retina was reattached with the SO tamponade, because the patient did not agree to remove the SO due to the pandemic. Two surgeries were required in 15 of the 19 eyes with an initial failure of reattachment, and three surgeries were required in four eyes. Multivariate analysis with initial failure of reattachment as the result of initial failure was higher in Group A and was statistically significant even when adjusted for age, sex, preoperative BCVA, and types of retinal breaks (odds ratio 3.35, 95% confidence interval 1.03–10.9, *p* = 0.04). Similarly, PVR was shown to be a factor in the initial failure of reattachment (odds ratio 5.88, 95% confidence interval 1.46–23.6, *p* = 0.01, Table 3).

The BCVA had improved significantly 6 months after the final surgery in both groups (*p* < 0.001 for Group A; *p* < 0.001 for Group B). The difference in the BCVA between the two groups at 6 months after the final surgery was not significant (Table 4). Revisions of the surgeries (except for recurrent RRDs) after a failure of the initial surgery were performed in four eyes in Group A and three eyes in Group B. One eye in Group A underwent a tube shunt surgery for refractory glaucoma, and the remaining six eyes had a subluxation of the implanted intraocular lens, and an intrascleral fixation of an intraocular lens was performed in all eyes, and the retina remained reattached at the last follow-up examination.

Because there were significant differences between the two groups in the sex distribution, incidence of PVR, and initial reattachment rate, we conducted subgroup analyses by dividing the two groups into those whose age was <55 years and those whose age was ≥55 years (Table 5). There was no difference in the sex distribution, frequency of PVR, and initial reattachment rate in the <55 years group for all factors, but there were significantly higher incidences of men, more PVR, and a lower initial reattachment rate in Group A for those aged ≥55 years.

## 4. Discussion

In 2020, the COVID-19 pandemic caused an unprecedented crisis worldwide, and with a lockdown in many countries, almost all outpatient services were canceled or postponed [11]. In the field of medical retina, there were reports of a 50% decrease in visits for intravitreal injections of anti-vascular endothelial growth factor (VEGF) drugs and a worsening of the disease [12]. The same crisis happened in surgical retina with several reports describing the results of hospital visits during the acute phase of RRD.

Patel et al. [13] reported that the examinations of the patients for RRD during the peak of COVID-19 pandemic in the USA were delayed with a more than two-fold increase in the chronicity of primary PVRs. They also reported that during the peak COVID-19 period, patients examined with RRD had poorer BCVAs and had more macular detachments by approximately 1.5-fold than that of the pre-pandemic period, which is in contrast with our results indicating a lack of increase in macular detachment during the pandemic peak. However, their younger patients (<50 years) did not show this disparity which was the same as our results. Similarly, Jasani et al. [14] reported that the incidence of macular detachment and PVR increased during the pandemic period in the United Kingdom.

Similar to the findings in our study, Ferreira et al. [15], from Portugal, found that there was a delay in being examined and an increase in the PVR with an increased use of silicone oil and C3F8 gas for the more complex RRD cases. This required more complex surgeries than conventional surgeries. In contrast, they found a higher rate of macular detachments at the time of presentation.

Interestingly, Moussa et al. [16], from United Kingdom, in their study to evaluate the effects of COVID-19 on RRD surgeries, compared a 9-month time period in the three successive pre-pandemic years to the pandemic year. They found that there was a decrease in the number of patients with an RRD, an increased number with macular involvement, with late presentation, and increased RRD primary failure during the first lockdown. These differences were not present during the second lockdown. They concluded that the attitudes of the patients, the health policies, and emergency services had adapted to the pandemic.

In Japan, a lockdown was not implemented and only a state of emergency declaration was made [17]. Then, the medical treatments were initially restricted and performed only at the discretion of each medical institution. Our hospital was required to implement surgical priorities of Grades A to C, and as in the past, only emergency surgeries with the highest priority of A could be performed. Standby surgeries classified as priority grade B or C were almost completely canceled and postponed. Although the total number of surgeries performed at our hospital was greatly reduced, surgeries for critical diseases including RRD were performed as before the pandemic, without any restrictions on visits and consultations while taking adequate infection control measures. Conventionally, the incidence of macular detachments in eyes with RRD at the initial visit is approximately 50%, and a Japanese multicenter study has shown similar findings [18]. In our institution, the incidence of a macular detachment in both groups was approximately 50% which was almost identical to that of a multicenter study [18]. Unlike previous reports [13,14,15,16], the incidence of macular detachment did not increase during the pandemic which may be because a lockdown was not instituted in Japan [17], and patients could receive medical care relatively easily compared to other countries. This resulted in earlier consultations by patients who were aware of vision reductions and visual field disturbances.

In spite of a slightly higher number of male patients in Group A (145 patients) compared to Group B (137 patients) during the pandemic, the decrease in hospital visits among women, especially those aged ≥ 55 years, was notable considering the 14% decrease in the total number of RRD surgeries. Because age is a risk factor for severe COVID-19 infections [19], it is relatively easy to assume that elderly patients may have refrained from visiting a doctor for fear of COVID-19 infection in hospitals. Actually, the patients in Group A were significantly younger than those in Group B in this study. It has been reported that the incidence of RRD in women ranged from 30.3% to 43.5% in the non-pandemic period [20]. In our cohort, the incidence of RRD in women was 27.5% in Group A and 40.7% in Group B which was clearly lower during the pandemic period. Hirakata et al. [21] reported a similar decrease in the number of RRD surgeries in Japan (Tokyo) among women during the pandemic, although the difference was not significant (*p* = 0.08). In addition to the cases of RRDs, a decrease in the rate of women receiving medical care for diabetes during the pandemic has been reported in Japan [22]. These findings suggest that women are generally more likely to avoid adverse risks than men in their response to health threats [23].

The decrease in the rate of examinations of women with RRD has not been observed in other countries [13,14,15,16]. Hirakata et al. [21] explained that the decrease in RRD among Japanese women was due to differences in the outdoor activities in women during emergencies. Hiroi reported that there was no sex difference in commuting in emergencies, and women were less likely to engage in activities for personal reasons such as shopping, social dining, socializing, and entertainment [24]. It was suggested that women may be more likely to refrain from outside activities due to self-control during the pandemic. In opposition, Patel et al. [13] did not find a male preponderance in their COVID-19 pandemic group. However, the duration of their study was short (50 days), and, hence, the follow-up period was short compared to the six-month study period in our study.

The number of PVR surgeries was significantly higher in Group A and especially in men. This suggests that it is more likely that men were the ones who ignored their vision reduction and refrained from seeing a doctor during the pandemic. We suggest that the decrease in RRD among women was not due to a decrease in the rate of medical examinations, but rather a decrease in the incidence of RRD among women due to the self-restraint in their behavior.

Patel et al. [13] reported that a macular detachment was less common in patients younger than 50 years of age and in patients who were being examined during their follow-up period. In our cohort, the incidence of macular detachment and PVR in Group A did not differ significantly from that in Group B in patients younger than 55 years, as reported [13]. This may be due to the lower incidence of mortality from COVID-19 infection in younger patients, who may have a lower threshold for visiting hospitals for medical treatments. In addition, patients who had a better knowledge of RRD during their regular follow-up visits to the hospital may have prevented severe ocular diseases, indicating that the education level of the patient was important during the pandemic.

In a meta-analysis, Roshanshad et al. [25] found that the COVID-19 lockdown was associated with a 53% to 66% reduction in RRD examinations, which was much higher than the 14% reported in this study. This was probably because there was no lockdown in Japan, which is supported by the increased macular detachment, increased duration of the symptoms, and increased PVR at presentation. They recommended that to reduce the costs of personal protective equipment that surgeons use phacovitrectomy instead of sequential surgery and develop more organized telemedicine to undertake surgical emergencies during a pandemic in the future.

This study has several limitations. This was a retrospective study conducted at a single university hospital, and thus patient selection bias was unavoidable due to restrictions on the healthcare system during the pandemic. Hence, a prospective multicenter study with a higher number of participants is required to overcome the deficiencies of the present study.

## 5. Conclusions

The number of RRD surgeries decreased by 14% during the COVID-19 pandemic in Japan. The PVR cases increased, especially in patients ≥55 years of age and the initial reattachment rate decreased. There was a significantly lower number of women who underwent RRD surgery, indicating that the epidemic has not subsided even in 2023, and the infection may be repeated in the future. The education of the patients and ophthalmic care must be continued.

## Figures and Tables

**Table 1 jcm-12-01522-t001:** Preoperative factors.

	Group A (203 Eyes)	Group B (235 Eyes)	*p*-Value
Age	52.5 ± 14.0	54.0 ± 14.1	0.04 ^†^
Sex (man/woman)	145/55	137/94	0.005 ^‡^
Preop BCVA (logMAR units)	0.51 ± 0.70	0.51 ± 0.68	0.65 ^†^
Posterior vitreous detachment	171 (84%)	197 (84%)	1 ^‡^
Macular detachment	96 (47%)	112 (48%)	1 ^‡^
Undetected retinal break	12 (6%)	6 (3%)	0.09 ^‡^
Numbers of retinal breaks	1.8 ± 0.8	1.7 ± 0.8	0.54 ^†^
Location of retinal breaks (upper/lower)	147/56	183/52	0.22 ^‡^
Size of retinal detachment (1–4 quadrants)	1.9 ± 1.0	1.9 ± 0.9	0.54 ^†^
PVR	17 (8%)	5 (2%)	0.004 ^‡^

Group A = pandemic group, Group B = control group, BCVA = best-corrected visual acuity, logMAR = logarithmic minimum angle of resolution, PVR = proliferative vitreoretinopathy, ^†^ Mann–Whitney U-test, ^‡^ Fisher’s exact probability test

**Table 2 jcm-12-01522-t002:** Background factors.

	Group A (203 Eyes)	Group B (235 Eyes)	*p*-Value ^‡^
Pseudophakia	32 (16%)	23 (10%)	0.06
Blunt trauma	10 (5%)	12 (5%)	1
High myopia	67 (33%)	65 (28%)	0.25
FEVR	3 (1%)	4 (2%)	1
Atopic dermatitis	9 (4%)	6 (3%)	0.30
Retinopathy of prematurity	2 (1%)	1 (0.4%)	0.60
Age ≥ 55-year-old	87 (43%)	137 (58%)	0.002

Group A = pandemic group, Group B = control group, FEVR = familial exudative vitreoretinopathy, ^‡^ Fisher’s exact probability test

**Table 3 jcm-12-01522-t003:** Multivariate analysis of odd ratio (OR) for the initial failure of reattachment.

	OR (95% CI)	*p*-Value *
Group A	3.35 (1.03–10.9)	0.04
Age	1.01 (0.98–1.04)	0.65
Sex (woman)	0.93 (0.30–2.94)	0.91
Preoperative BCVA	1.10 (0.47–2.60)	0.82
Numbers of retinal breaks	0.94 (0.50–1.75)	0.83
Location of retinal breaks (lower)	2.25 (0.81–6.29)	0.12
Area of retinal detachment	1.34 (0.73–2.47)	0.34
PVR	5.88 (1.46–23.6)	0.01

CI = confidence interval, Group A = pandemic group, BCVA = best-corrected visual acuity, PVR = proliferative vitreoretinopathy, * Logistic regression analysis

**Table 4 jcm-12-01522-t004:** Preoperative and postoperative best-corrected visual acuity in logMAR units.

	Group A	Group B	*p*-Value
Preoperative BCVA (logMAR)	0.51 ± 0.70	0.51 ± 0.68	0.65 ^†^
Postoperative BCVA * (logMAR)	0.06 ± 0.25	0.05 ± 0.25	0.65 ^†^
*p*-value	<0.001 **	<0.001 **	

Group A = pandemic group; Group B, control group, BCVA = best-corrected visual acuity, logMAR: logarithmic minimum angle of resolution, Postoperative BCVA * determined at 6 months after the last surgery, ^†^ Mann–Whitney U-test, ** Wilcoxon signed rank test.

**Table 5 jcm-12-01522-t005:** Subgroup analysis.

	Group A (203 Eyes)	Group B (235 Eyes)	*p*-Value ^‡^
Sex (man/woman); age < 55 years	80/36	59/39	0.20
Sex (man/woman); age ≥ 55 years	66/21	80/57	0.009
PVR; age < 55 years	8 (7%)	4 (4%)	0.55
PVR; age ≥ 55 years	9 (10%)	1 (0.7%)	0.001
Initial reattachment rate; age < 55 years	108 (93%)	96 (98%)	0.11
Initial reattachment rate; age ≥ 55 years	80 (92%)	135 (99%)	0.003

Group A = pandemic group, Group B = control group, PVR = proliferative vitreoretinopathy, ^‡^ Fisher’s exact probability test.

## Data Availability

The data presented in this study are available on request from the first author (M.M.).

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
