# Peer review of "Effect of the COVID-19 Pandemic on Surgical Outcomes for Rhegmatogenous Retinal Detachments"

_jcm, 2023, doi:10.3390/jcm12041522_

Round 1

Reviewer 1 Report (Previous Reviewer 1)

This is a great work comparing retinal detachment in COVID and no-COVID period. The text is well organized and thought are fine.

I was not expecting about macular detachment being similar between both groups when PVR is increased in the COVD period and also silicoin oil use.

Author Response

Answers for Reviewer 1:

We thank you and the reviewers for the comments to our manuscript. The manuscript has been revised according to the comments, and our answers are included.

This is a great work comparing retinal detachment in COVID and no-COVID period. The text is well organized and thought are fine.

I was not expecting about macular detachment being similar between both groups when PVR is increased in the COVD period and also silicon oil use.

Answer: We thank you for your comments. The following sentences were inserted in line 236, “Similar to the findings in our study, Ferreira et al. [15] from Portugal found that there was a delay in being examined and an increase in the PVR with increased use of silicone oil and C3F8 gas for the more complex RRD cases. This required more complex surgeries than the conventional surgeries. In contrast, they found a higher rate of macular detachments at the time of presentation.”

Reviewer 2 Report (Previous Reviewer 2)

The authors need to be commended for their efforts to improve their study by increasing the time period from 3 months to 6 months and the number of patients from 203 patients to 431 patients. Please find below a few comments.

.

As in the present study, Patel et al. from the US showed that RRD during COVID-19 peak had delayed presentation, and increased chronicity with primary PVR.1 Also, they found that compared to the pre-pandemic time period, during peak COVID, overall, patients had worse logMAR VA and more macular involvement, though their cohort of younger patients (<50 years) were spared of this disparity, in contrast to the present study. As per the authors of the present study, lack of increase of macular detachment during the pandemic peak in their study was because in contrast with other nations, lockdown was not instituted in Japan, and hence patients had easy access to medical care. According to Patel et al. this could be explained by the lower threshold for younger patients to seek emergency medical help during the pandemic peak due to perception of lower morbidity and mortality. Additionally, Patel et al. reported that established patients too seemed to be spared of this pandemic induced disadvantage. As opposed to the findings in the present study, Patel et al did not find male preponderance,  in their COVID group. The authors of the present study ascribe this to better self-control during the pandemic-imposed restrictions in women compared to men. However, the duration of Patel et al. study was short (50 days), and hence follow up period was limited, compared to the six months study period in the present study.

Roshanshad et al., in their meta-analyses, found that COVID-19 lockdown was associated with 53-66% reduction in RD presentation (much higher than the 14% reported in the present study, probably again because of no lockdown in Japan), increased macular involvement, increased duration of symptoms and increased PVR at presentation.2 They recommended that to cut down costs of PPE, etc. to utilize phacovitrectomy instead of sequential surgery and develop more organized telemedicine to tackle such pandemics in the future.

Like the present study, Ferreira et al. from Portugal, in their longer study period of 1 year, also found delayed presentation and increased PVR (increased usage of silicone oil and C3F8 gas for more complex cases requiring more complex surgeries was used as a surrogate for this).3 In contrast, they also found higher rate of macular detachment at presentation.

Interestingly, Moussa et al. from UK, in their very recent study to evaluate the sustained effect of COVID-19 on RRD repair, compared a 9-month time period in the 3 successive years pre-pandemic to the pandemic year.4 They found that though during the first lockdown, there was decreased number of patients with RRD, increased macular involvement, late presentation and increased RRD primary failure, these differences ceased to exist during the second lockdown. They concluded that patient attitudes, health policies and emergency services had adapted to the pandemic.

1.     Patel LG, Peck T, Starr MR, Ammar MJ, Khan MA, Yonekawa Y, Klufas MA, Regillo CD, Ho AC, Xu D. Clinical Presentation of Rhegmatogenous Retinal Detachment during the COVID-19 Pandemic: A Historical Cohort Study. Ophthalmology. 2021 May;128(5):686-692. doi: 10.1016/j.ophtha.2020.10.009. Epub 2020 Oct 13. PMID: 33058938; PMCID: PMC7550253.

2.     Roshanshad A, Binder S. Retinal detachment during COVID-19 era: a review of challenges and solutions. Spektrum Augenheilkd. 2022;36(1):32-37. doi: 10.1007/s00717-021-00493-7. Epub 2021 Jun 30. PMID: 34226798; PMCID: PMC8243622.

3.     Ferreira A, Afonso M, Silva N, Meireles A. The Impact of COVID-19 Pandemic on Surgical Primary Retinal Detachments. Ophthalmologica. 2022;245(2):111-116. doi: 10.1159/000520342. Epub 2021 Oct 21. PMID: 34673635; PMCID: PMC8678219.

4.   Moussa G, Qadir MO, Ch'ng SW, Lett KS, Mitra A, Tyagi AK, Sharma A, Andreatta W. Sustained impact of COVID-19 on primary retinal detachment repair in a tertiary eye hospital from March to December 2020. Spektrum Augenheilkd. 2022 May 24:1-8. doi: 10.1007/s00717-022-00521-0. Epub ahead of print. PMID: 35645464; PMCID: PMC9127495.

Author Response

Answers for Reviewer 2:

We thank you and the reviewers for the comments to our manuscript. The manuscript has been revised according to the comments, and our answers are included.

The authors need to be commended for their efforts to improve their study by increasing the time period from 3 months to 6 months and the number of patients from 203 patients to 431 patients. Please find below a few comments.

  1. As in the present study, Patel et al. from the US showed that RRD during COVID-19 peak had delayed presentation, and increased chronicity with primary PVR.1Also, they found that compared to the pre-pandemic time period, during peak COVID, overall, patients had worse logMAR VA and more macular involvement, though their cohort of younger patients (<50 years) were spared of this disparity, in contrast to the present study. As per the authors of the present study, lack of increase of macular detachment during the pandemic peak in their study was because in contrast with other nations, lockdown was not instituted in Japan, and hence patients had easy access to medical care. According to Patel et al. this could be explained by the lower threshold for younger patients to seek emergency medical help during the pandemic peak due to perception of lower morbidity and mortality. Additionally, Patel et al. reported that established patients too seemed to be spared of this pandemic induced disadvantage. As opposed to the findings in the present study, Patel et al did not find male preponderance, in their COVID group. The authors of the present study ascribe this to better self-control during the pandemic-imposed restrictions in women compared to men. However, the duration of Patel et al. study was short (50 days), and hence follow up period was limited, compared to the six months study period in the present study.

Answer: We thank you for your comments. The following sentences were inserted in line 226, “Patel et al. [13] reported that the examinations of the patients for RRD during the peak of COVID-19 pandemic in the USA were delayed with an increase of chronicity of primary PVRs by more than 2-fold. They also reported that during the peak COVID-19 period, the patients with RRD examined had poorer BCVAs and had more macular detachments by approximately 1.5-fold than that of the pre-pandemic period which is in contrast with our results indicating a lack of increase of macular detachment during the pandemic peak. However, their younger patients (<50 years) did not show this disparity which was the same as our results. Similarly, Jasani et al. [14] reported that the incidence of macular detachment and PVR increased during the pandemic period in the United Kingdom. Similarly, Jasani et al. [14] from the United Kingdom reported that the incidences of macular detachment and PVR increased during the pandemic period.”

In line 286, “In opposition, Patel et al. [13] did not find a male preponderance in their COVID-19 pandemic group. However, the duration of their study was short (50 days), and hence the follow-up period was short compared to the six months study period in our study.”

  1. Roshanshad et al., in their meta-analyses, found that COVID-19 lockdown was associated with 53-66% reduction in RD presentation (much higher than the 14% reported in the present study, probably again because of no lockdown in Japan), increased macular involvement, increased duration of symptoms and increased PVR at presentation.2They recommended that to cut down costs of PPE, etc. to utilize phacovitrectomy instead of sequential surgery and develop more organized telemedicine to tackle such pandemics in the future.

Answer: The following sentences and additional reference were inserted in line 305, “In a meta-analysis, Roshanshad et al. [25] found that COVID-19 lockdown was associated with a 53% to 66% reduction in RRD examinations which was much higher than the 14% reported in this study. This was probably because there was no lockdown in Japan which is supported by the increased macular detachment, increased duration of the symptoms, and increased PVR at presentation. They recommended that to reduce the costs of personal protective equipment that surgeons use phacovitrectomy instead of sequential surgery and develop more organized telemedicine to undertake surgical emergencies during a pandemic in the future.

  1. Like the present study, Ferreira et al. from Portugal, in their longer study period of 1 year, also found delayed presentation and increased PVR (increased usage of silicone oil and C3F8 gas for more complex cases requiring more complex surgeries was used as a surrogate for this).3In contrast, they also found higher rate of macular detachment at presentation.

Answer: We thank you for your comments. The following sentences were inserted in line 236, “Similar to the findings in our study, Ferreira et al. [15] from Portugal found that there was a delay in being examined and an increase in the PVR with increased use of silicone oil and C3F8 gas for the more complex RRD cases. This required more complex surgeries than the conventional surgeries. In contrast, they found a higher rate of macular detachments at the time of presentation.”

  1. Interestingly, Moussa et al. from UK, in their very recent study to evaluate the sustained effect of COVID-19 on RRD repair, compared a 9-month time period in the 3 successive years pre-pandemic to the pandemic year.4They found that though during the first lockdown, there was decreased number of patients with RRD, increased macular involvement, late presentation and increased RRD primary failure, these differences ceased to exist during the second lockdown. They concluded that patient attitudes, health policies and emergency services had adapted to the pandemic.

Answer: The following sentences and additional reference were inserted in line 241, “Interestingly, Moussa et al. [16] from United Kingdom, in their study to evaluate the effects of COVID-19 on RRD surgeries, compared a 9-month time period in the 3 successive pre-pandemic years to the pandemic year. They found that there was a decrease in the number of patients with an RRD, increased number with macular involvement, with late presentation, and increased RRD primary failure during the first lockdown. These differences were not present during the second lockdown. They concluded that the attitudes of the patients, the health policies, and emergency services had adapted to the pandemic.

New reference numbers

  1. Patel LG, Peck T, Starr MR, Ammar MJ, Khan MA, Yonekawa Y, Klufas MA, Regillo CD, Ho AC, Xu D. Clinical Presentation of Rhegmatogenous Retinal Detachment during the COVID-19 Pandemic: A Historical Cohort Study. Ophthalmology. 2021 May;128(5):686-692. doi: 10.1016/j.ophtha.2020.10.009. Epub 2020 Oct 13. PMID: 33058938; PMCID: PMC7550253.

  1. Roshanshad A, Binder S. Retinal detachment during COVID-19 era: a review of challenges and solutions. Spektrum Augenheilkd. 2022;36(1):32-37. doi: 10.1007/s00717-021-00493-7. Epub 2021 Jun 30. PMID: 34226798; PMCID: PMC8243622.

  1. Ferreira A, Afonso M, Silva N, Meireles A. The Impact of COVID-19 Pandemic on Surgical Primary Retinal Detachments. Ophthalmologica. 2022;245(2):111-116. doi: 10.1159/000520342. Epub 2021 Oct 21. PMID: 34673635; PMCID: PMC8678219.

  1.  Moussa G, Qadir MO, Ch'ng SW, Lett KS, Mitra A, Tyagi AK, Sharma A, Andreatta W. Sustained impact of COVID-19 on primary retinal detachment repair in a tertiary eye hospital from March to December 2020. Spektrum Augenheilkd. 2022 May 24:1-8. doi: 10.1007/s00717-022-00521-0. Epub ahead of print. PMID: 35645464; PMCID: PMC9127495.

This manuscript is a resubmission of an earlier submission. The following is a list of the peer review reports and author responses from that submission.

Round 1

Reviewer 1 Report

This is an interesting work comparing results and preoperative features of RRD during pandemic period versus a non-pandemic one. This is important because health access was not so easy but otherwise behavior was also better because people were mostly at home.

Methods

Surgical Procedures:

“If the SO was not removed, the eye 82 was assumed to be still detached.” – why? This will influence the attachment results because the operative room access was not so easy during the pandemic time and SO removal could me extended but retina could be attached.

Results:

Table 4: Are you sure about BCVA in logMAR as number 15 and 4?

Author Response

We thank you and the reviewers for the comments to our manuscript. The manuscript has been revised according to the comments, and our answers are included.

1: This is an interesting work comparing results and preoperative features of RRD during pandemic period versus a non-pandemic one. This is important because health access was not so easy but otherwise behavior was also better because people were mostly at home.

Answer: We thank you.

2: Methods

Surgical Procedures:

“If the SO was not removed, the eye 82 was assumed to be still detached.” – why? This will influence the attachment results because the operative room access was not so easy during the pandemic time and SO removal could be extended but retina could be attached.

Answer: The following sentences were inserted in line 158:

“SO tamponade was used in 2 eyes in Group A and 2 eyes in Group B. The SO was removed in 3 eyes within 6 months and the retina remained attached. The one eye in Group A that was not reattached was a case in which the SO was not removed at 6 months after the last surgery even though the retina was reattached with the SO tamponade, but the patient did not agree to remove SO due to the pandemic.”

3: Results:

Table 4: Are you sure about BCVA in logMAR as number 15 and 4?

Answer: We apologize for the mistakes in Table 4, and they have been corrected.

Group A

Group B

P-value

Preoperative BCVA (logMAR)

0.47±0.65

0.49±0.65

0.56

Postoperative BCVA* (logMAR)

0.06±0.22

0.07±0.28

0.94

P-value

<0.001**

<0.001**

Reviewer 2 Report

The authors have analysed a very important and relevant subject.

They found that for those >50 years of age during the COVID pandemic lockdown in Japan, more men than women, greater PVR and lower initial retinal settlement rate were seen when compared with a similar time period in the year before the COVID pandemic.

Unlike previous reports they did not find an increase in macular detachment during the pandemic. They hypothesise that women were more careful in curtailing their outdoor activities during the pandemic and probably this resulted in their lower incidence of RRD.  Also men neglected their symptoms of RRD, hence had a higher incidence of PVR.

The authors have mentioned the limitations of their study: retrospective design, a single centre study and patient selection bias

Hence a prospective multi-centre study with a higher number of participants is required to overcome the lacunae of the present study

Author Response

We thank you for the comments of our manuscript. The manuscript has been revised according to the comments, and our answers are included.

The authors have analyzed a very important and relevant subject.

They found that for those >50 years of age during the COVID pandemic lockdown in Japan, more men than women, greater PVR and lower initial retinal settlement rate were seen when compared with a similar time period in the year before the COVID pandemic.

Unlike previous reports they did not find an increase in macular detachment during the pandemic. They hypothesize that women were more careful in curtailing their outdoor activities during the pandemic and probably this resulted in their lower incidence of RRD. Also, men neglected their symptoms of RRD, hence had a higher incidence of PVR.

The authors have mentioned the limitations of their study: retrospective design, a single centre study and patient selection bias.

Hence a prospective multi-centre study with a higher number of participants is required to overcome the lacunae of the present study

Answer: We thank you for your comments. The following sentences were inserted in line 269,

“Hence a prospective multi-center study with a higher number of participants is required to overcome the voids of the present study.”
